# Patient specific Pancreatic Ductal Adenocarcinoma segmentation in multiphase CTs through a registration methodology

Sara Ghezzi[1], Edoardo Maria Polo[1], Riccardo Levi[2,5], Cristian Drudi[1], Rami Abou El Zahab[4], Silvia Carrara[3,5], Cristiana Bonifacio[2], Alessandro Zerbi[4,5], and Riccardo Barbieri[1] *Senior Member, IEEE*

*Abstract*—Dual-phase computed tomography, comprising late arterial (LA) and portal venous (PV) phases, is essential for Pancreatic Ductal Adenocarcinoma (PDAC) assessment. However, automated PDAC detection remains challenging due to phase-dependent contrast variations. We present a patient-specific paradigm using clinically-validated LA phase segmentation and inter-phase registration to align PV phase segmentations with the corrected LA phase, enabling more accurate PV phase annotation. In 21 PDAC patients, we observed consistent automated LA segmentation of single-mass tumors, while 9.5% of automated PV phase segmentations exhibited disconnected regions. Our registration-based approach achieved a median Dice score of 0.85 for pancreas segmentation, significantly improving upon the unregistered PV phase performance of 0.79 and approaching the automated LA phase performance of 0.90. Furthermore, it demonstrated comparable centroid distance accuracy to automated LA segmentation (p>0.05). This approach enables efficient multi-phase PDAC analysis by only requiring manual correction in the optimal LA phase. Our methodology addresses phase-dependent segmentation challenges while optimizing clinical workflow, potentially improving diagnostic efficiency and treatment planning

*Index Terms*—Pancreatic Ductal Adenocarcinoma; Deep Learning; Multi-phase CT; Image Registration; Tumor Segmentation; nnU-Net

## I. INTRODUCTION

Pancreatic ductal adenocarcinoma (PDAC) represents the most common form of pancreatic cancer, with a dismal 10.8% 5-year relative survival rate. The poor prognosis is predominantly attributable to late detection, with 80-85% of patients exhibiting as unresectable or metastatic at diagnosis [1]. Although Magnetic Resonance Imaging demonstrates comparable sensitivity in local staging assessement, Computed Tomography (CT) remains the imaging modality of choice due to its superior spatial resolution, rapid acquisition, and broader familiarity among multidisciplinary clinicians [2]. Recent advances in deep learning have facilitated automated

[1] Politecnico di Milano, Piazza Leonardo da Vinci, 32, Milan, 20133, Italy
[2] Radiology Department, IRCCS Humanitas Research Hospital, Via Manzoni 56, 20089 Rozzano, Milan, Italy
[3] Gastroenterology Department, Endoscopic Unit, IRCCS, Humanitas Research Hospital, Rozzano, Milan, Italy
[4] Pancreatic Surgery Unit, IRCCS Humanitas Research Hospital, Rozzano, MI, Italy
[5] Department of Biomedical Sciences, Humanitas University, Pieve Emanuele, MI, Italy

PDAC detection in CT imaging [3], offering promising opportunities for early-stage diagnosis assistance. Automated tumor segmentation represents a crucial development in enhancing measurement efficiency and biomarker reproducibility compared to manual delineation [4]. In this context, the nnU-Net architecture [5] established itself as the state-of-the-art standard for organ and tumor segmentation. While demonstrating exceptional performance with Dice Similarity Coefficients (DSC) exceeding 97% for organs such as liver, kidney, and spleen [6], pancreatic segmentation performance remains suboptimal at approximately 80% [7], with PDAC detection rates reaching 72% [16]. The National Comprehensive Cancer Network guidelines establish dual-phase pancreatic CT protocol as essential for comprehensive tumor evaluation, vascular involvement assessment, and metastasis detection [2]. This dual-phase protocol, encompassing late arterial (LA) and portal venous (PV) phases, represents the primary imaging approach for suspected pancreatic malignancies. PDAC typically manifests as a hypoattenuating lesion during the LA phase, which optimizes visualization of arterial involvement, anatomical variants, and stenoses—crucial elements for treatment planning. Complementarily, the PV phase facilitates optimal visualization and assessment of the portomesenteric venous system, which may be affected by primary tumor invasion or metastatic involvement [8] [9]. Multi-phase imaging enhances tumor-stroma ratio prediction accuracy, with feature fusion modules demonstrating significant performance improvements [10]. Studies show that nnU-Net architectures trained on a concatenated multi-phase CT input data achieve superior PDAC segmentation performance metrics compared to single-phase methodologies [4]. However, tumor texture variations and patient movement between phases continue to pose a significant challenge [11], frequently necessitating expert clinical review and manual segmentation refinement. Moreover, existing approaches predominantly focus on generating singular optimized segmentations for the venous phase only. Given the importance of employing multiple phases, methodologies addressing segmentation accuracy across all acquisition phases remain underdeveloped.

We propose an efficient methodology leveraging clinically validated pancreas segmentations from the LA phase to enable accurate tumor localization across phases. Our approach focuses on the registration of the PV phase pancreas seg-

mentations to expert-corrected LA phase segmentations. Since PDAC typically originates and develops within the pancreatic parenchyma, achieving accurate pancreas registration between phases provides a reliable framework for tumor annotation transfer. This strategy is particularly valuable as PDAC visualization is optimal in the LA phase, where clinicians can more accurately delineate tumor boundaries compared to the reduced contrast in PV phase. Through implementation of accurate pancreatic parenchymal alignment between phases, our methodology establishes a reliable framework for tumor annotation transfer from LA to alternative phases, including PV, while preserving anatomical fidelity and spatial relationships.

## II. MATERIALS AND METHODS

### A. Patients Data

Abdominal CT scans were acquired from 21 patients diagnosed with PDAC at Humanitas Reasearch Hospital using different CT scanners. For each patient, two distinct volumes were obtained, corresponding to different contrast phases: LA and PV. Segmentation was provided exclusively for the LA phase. The protocol was approved by Humanitas ethical committee (204/23). The parameters, including spacing and size, were not uniform across volumes from the same patient due to varying clinical necessities. Pancreatic and PDAC segmentations, corrected by radiologists from Humanitas Reasearch Hospital, were provided exclusively for the LA phase. Patients enrolled in this study were required to be at least 18 years old, not pregnant, and capable of expressing informed consent. In addition to the aforementioned volumes, a pancreatic dataset of the Medical Segmentation Decathlon (MSD) [12] was utilized to train a Convolutional Neural Network. This dataset comprises 281 abdominal CT volumes of PV phase acquired at the Memorial Sloan Kettering Cancer Center (Manhattan, NY, USA). The dataset is made available together with segmentations of pancreas and pancreatic malignancies, including PDAC.
All images used are anonymized and in NIFTI format.

### B. nnU-Net and tumour assesment

Prior to image registration, a neural network with a U-Net-like architecture was trained following the deep learning-based segmentation method known as no-new U-Net (nnU-Net), developed by Isensee et. al [5]. The network was trained on 80% of the MSD pancreas dataset using an NVIDIA GPU RTX 3060, while the remaining portion of the dataset was reserved for validation. Since nnU-Net is considered the state-of-the art for medical segmentation [1], it was then applied to the 21 acquired CT volumes to generate gold-standard automatic segmentations of pancreas and tumor both for the LA and PV phase. These segmentations were further utilized during the registration process, as detailed below.
Using the LA phase as baseline, we calculated Dice similarity coefficients for both pancreatic parenchyma and tumor segmentations between network outputs and clinically-corrected ground truth. Given the documented challenges in automated tumor segmentation, we anticipated lower Dice scores for tumor compared to parenchymal tissue. To provide detailed analysis of tumor segmentation quality, we evaluated the number of unconnected regions in nnU-Net segmentations for both LA and PV phases, given that clinically-corrected LA phase segmentations consistently presented tumors as single connected masses. Connected components are defined as groups of tumor voxels that maintain continuity through direct or diagonal adjacency, while regions separated by background voxels are identified as disconnected components.

### C. Registration algorithm

The algorithm was developed according to the pipeline presented in this section.

- **Volumes pre-processing:**
  Initially, volumes were processed to facilitate smoother and more robust convergence. To this purpose, pancreas segmentations obtained using nnU-Net were employed to reduce the z-axis dimensions by retaining only the slices that contained the pancreas. A tolerance of 6 slices was applied, including 3 slices before and 3 after the organ. Voxel intensities were adjusted using a window center of 40 and a window width of 350, i.e., values below -155 Hounsfield Units (HU) were set to 0, while those above +195 HU were capped at 255. Furthermore, PV phase volumes with different voxel spacings relative to their corresponding LA phase were resampled to ensure consistency.

- **Global registration:**
  A global registration was performed using the entire patient body as reference. At each iteration, the similarity metric was computed within a binary mask encompassing the patient's body volume, thereby excluding background regions and the scanner table, which remained invariant across all acquired volumes. An affine transformation was employed as the initial transformation step to address global translational and rotational misalignments between phases. This preliminary affine registration established a coarse alignment baseline, significantly reducing computational burden and convergence time for subsequent elastic registration.

- **Focused Pancreas registration:**
  A refined registration specifically targeting the pancreatic region was then performed to achieve optimal pancreas alignment. To avoid bias, only the LA segmentation obtained with nnU-Net was used to define the pancreatic region, since registration evaluation relied on comparing the transformed PV phase pancreas segmentation against the ground truth LA segmentation. For the Focused Pancreas registration, two different transformation approaches were evaluated and combined to first step: rigid (affine) transformation and elastic (B-spline) transformation. The elastic approach allows for non-linear deformations that can better accommodate local tissue variations and organ deformation between phases, potentially improving registration accuracy in cases where rigid

transformation is insufficient.

To comprehensively evaluate registration strategies, we also assessed elastic transformation applied directly to the pancreatic region without prior rigid alignment of the entire volumes.

- **Transfer of PV segmentations to LA:** Using the final transformation achieved in the previous step, pancreatic segmentations of the PV phase were transposed to LA phase.

Both registrations employed the Mutual Information (MI) metric proposed by Mattes et al., with marginal and joint probability density functions constructed at discrete positions (bins). The number of bins was optimized using grid search, selecting the value that maximized the Dice Similarity Coefficient (DSC) between segmentations for each volume pair. During global registration, 90% of voxels were sampled for metric calculation, while focused registration used 75% sampling due to the smaller mask dimensions. In addition to classical MI, in the focused registration step, we evaluated a weighted Mutual Information to enhance the algorithm's sensitivity to different pixel intensities across CT phases. Six distinct approaches that combine these transformation types with different initialization strategies and similarity metrics.

Gradient descent was employed as the optimizer using a learning rate of 1. Iterations continued until either the limit of 300 epochs was reached or the change in the metric between two consecutive steps was smaller than $10^{-5}$. At the end of each iteration, the moving image was interpolated using linear interpolation to resample voxels positioned at non-integer coordinates. For each patient, all six methods were evaluated with parameter optimization, and the best-performing method was selected based on DSC scores, enabling a patient-specific registration approach.

The six distinct registration strategies were implemented by combining the Global and the Focused Pancreas registration methods:

### Standard Methods:

- M1: Rigid Global $\rightarrow$ Rigid Focused Pancreas (two-step rigid registration)
- M2: Rigid Global $\rightarrow$ Elastic Focused Pancreas (global rigid + local elastic)
- M3: Elastic Focused Pancreas (single-step elastic registration)

### Phase-Aware Methods:

- M4: Rigid Global $\rightarrow$ Rigid Focused Pancreas + Weighted MI
- M5: Rigid Global $\rightarrow$ Elastic Focused Pancreas + Weighted MI
- M6: Elastic Focused Pancreas + Weighted MI

The phase-aware methods leverage the distinct enhancement characteristics of pancreatic structures in multi-phase CT. Based on established radiological criteria, tissues are classified by their enhancement patterns: hypovascular regions consistent with PDAC ($<20$ HU enhancement), arterial structures (30-100 HU enhancement), and normal parenchyma (20-50

HU enhancement). These classifications are based on well-established literature: PDAC is notoriously hypovascular with typical enhancement of 10-20 HU [17], while arterial structures show strong and predictable enhancement serving as stable anatomical landmarks [18], and normal pancreatic parenchyma exhibits physiological enhancement of 20-50 HU [19]. These classifications guide registration by prioritizing clinically relevant anatomical landmarks through weighted MI metrics. Figure 1 shows the flowchart of the study.

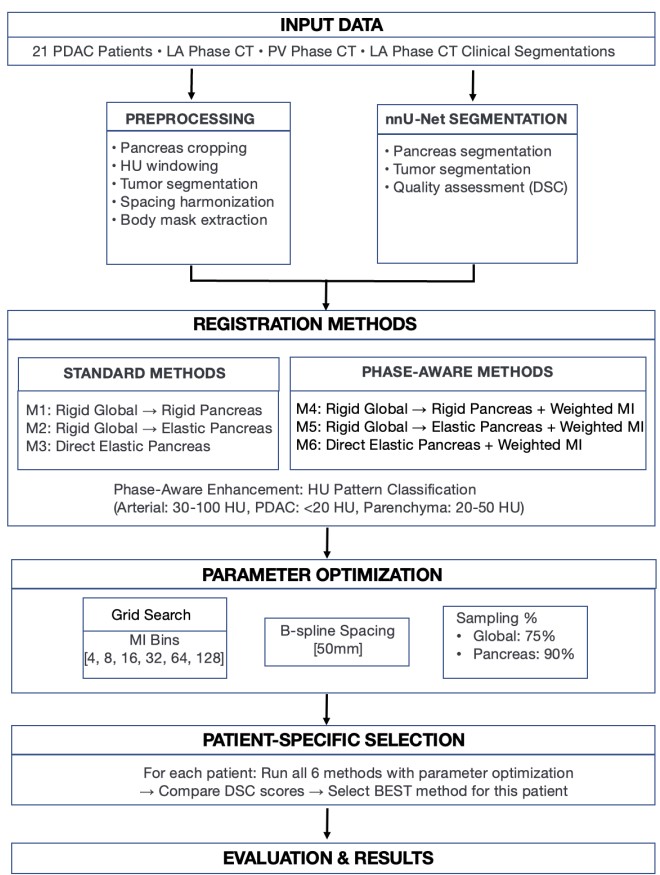

Fig. 1. Workflow for multi-phase CT registration in 21 PDAC patients. After preprocessing and nnU-Net segmentation, six registration methods are compared: standard approaches (M1-M3) and phase-aware methods (M4-M6) using weighted MI based on tissue enhancement patterns. Parameter optimization and patient-specific method selection enable personalized registration strategies based on DSC performance.

### D. Statistical analysis

Two quantitative metrics were used to evaluate pancreatic segmentation accuracy: centroid distances and DSC. These metrics were calculated between clinically-corrected LA phase segmentations (serving as the reference standard) and three different segmentation approaches: nnU-Net on LA phase, nnU-Net on PV phase, and registered PV segmentations. Given the non-normal distribution of both metrics, statistical comparisons were performed using the Wilcoxon signed-rank test, with statistical significance set at $p < 0.05$. To assess

the performance of the PV-to-LA registration algorithm, both the nnU-Net PV vs. corrected LA and registered PV vs. corrected LA comparisons were evaluated against the baseline performance (nnU-Net LA vs. corrected LA segmentations).

## III. RESULTS

We performed quantitative assessment of segmentation performance across different contrast-enhanced CT phases and anatomical structures, as detailed in Table I. The nnU-Net performance on LA phase demonstrated superior accuracy for both pancreatic parenchyma and PDAC compared to PV phase segmentations. Notably, the lower accuracy in PDAC segmentation across all approaches suggests a greater need for manual refinement compared to pancreatic parenchyma segmentation. Connectivity analysis revealed that both clinically-corrected LA and nnU-Net LA outputs consistently maintained single connected tumor regions for each subject. In contrast, nnU-Net PV segmentations showed discontinuities, with two subjects presenting disconnected tumor regions. To address these limitations, we implemented a registration approach for PV phase segmentations. The patient-specific selection of the optimal registration method (based on highest DSC score) achieved substantial improvement over direct PV phase segmentation, as shown in Table I. While we report PDAC metrics for all approaches for completeness, our registration strategy primarily focuses on achieving accurate pancreatic parenchyma alignment between phases. Our pancreas-centric registration approach is motivated by the superior automated segmentation performance of pancreatic parenchyma compared to tumor tissue, enabling reliable transfer of clinically-corrected tumor annotations from the optimal LA phase to PV phase. This is because precise pancreas registration provides the anatomical framework necessary for tumor annotation transfer from LA to PV phase, where PDAC visualization and delineation are optimal in the LA phase.

Statistical analysis using Wilcoxon signed-rank test revealed significant differences in pancreas centroid distances (relative to clinically-corrected LA segmentations) between nnU-Net PV vs. nnU-Net LA (p=0.035), while no significant difference was found between registered PV vs. nnU-Net LA (p=0.76). Similarly, DSC analysis against clinically-corrected LA segmentations showed significant differences in both comparisons (registered PV vs. nnU-Net LA and nnU-Net PV vs. nnU-Net LA). These distributions and their statistical significance are illustrated in Figure 2.

The effect of the PV to LA registration is visualized in Figure 3 for an example subject. The original PV segmentation (blue) exhibits notable misalignment with the LA ground truth (green). After registration (red), the segmentation closely matches the ground truth, highlighting the effectiveness of the registration approach in aligning the PV segmentation to the LA phase anatomy while maintaining the integrity of the pancreas and tumor segmentation masks.

The patient-specific method selection revealed interesting patterns in registration performance across our cohort of 21 PDAC patients as shown in Figure 4. The distribution

TABLE I
QUANTITATIVE EVALUATION OF PANCREAS AND PDAC
SEGMENTATION PERFORMANCE

| Method | DSC (median±MAD) | | Centroid Distance (voxels) | |
|---|---|---|---|---|
| | Pancreas | PDAC | Pancreas | PDAC |
| nnU-Net LA | 0.90±0.03 | 0.76±0.12 | 1.9±2.4 | 3.2±3.3 |
| nnU-Net PV | 0.79±0.07 | 0.53±0.15 | 4.1±1.5 | 6.29±2.13 |
| Registered PV | 0.85±0.05 | 0.66±0.18 | 2.8±1.3 | 4.35±2.34 |

DSC: Dice Similarity Coefficient; MAD: Median Absolute Deviation
All metrics are computed against clinically-corrected LA phase segmentations as ground truth
LA: Late Arterial; PV: Portal Venous; PDAC: Pancreatic Ductal Adenocarcinoma

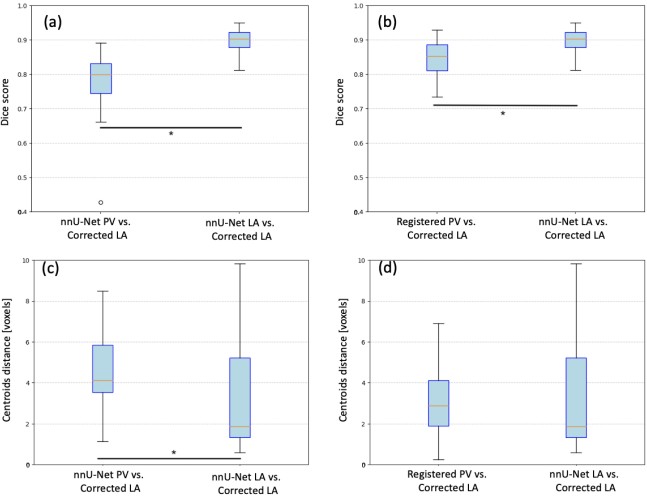

Fig. 2. Analysis of segmentation accuracy: above, the distributions of DSC comparing nnU-Net LA with nnU-Net PV (a) or with registered PV (b); below, the distrbutions of centroid distances between nnU-Net LA vs. nnU-Net PV (c) and nnU-Net LA vs. registered PV. Statistically significant differences are indicated by asterisks (*).

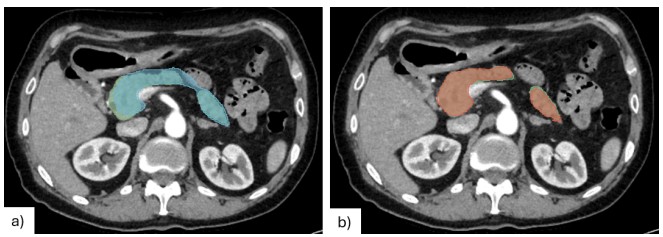

Fig. 3. Comparison between PV pancreatic segmentation and its transformed version to LA segmentation. On the left (a): PV segmentation (blue) compared to ground truth LA segmentation (green). On the right (b): registered PV segmentation (red) compared to ground truth (green).

shows that Elastic Focused Pancreas method with phase-aware enhancement performed best in the largest number of patients (7/21, 33.3%), followed by Rigid Global + Rigid Focused Pancreas with phase-aware enhancement (5/21, 23.8%) and only Elastic Focused Pancreas (5/21, 23.8%), while standard Rigid Global + Rigid Focused Pancreas was optimal in 4 patients (19.0%). Notably, none of the patients achieved optimal results with the multi-step rigid-to-elastic approach (Methods M2 and M5).

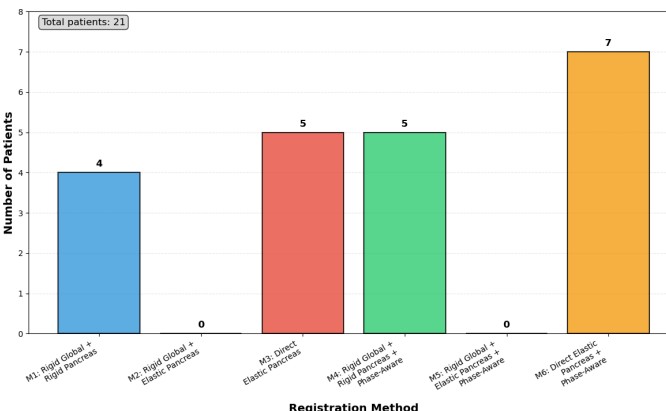

Fig. 4. Histogram showing the number of patients for which each registration method (M1-M6) achieved the highest DSC score. Multi-step approaches (M2: Rigid Global → Elastic Pancreas; M5: Rigid Global → Elastic Pancreas + Phase-Aware) were never selected as optimal, while direct elastic registration with phase-aware enhancement (M6) performed best in the largest subset of patients (n=7, 33.3%).

## IV. DISCUSSION

Analysis of DSC between nnU-Net and clinically-corrected segmentations in the LA phase revealed segmentation discrepancies, with median scores of 0.90 and 0.76 for pancreas and tumor, respectively (both < 1.0). The lower DSC for PDAC segmentation aligns with known challenges in tumor delineation compared to pancreatic parenchyma, despite the enhanced tumor contrast characteristic of LA phase imaging [9] [11].
Connected component analysis revealed distinct tumor segmentation patterns across phases. While nnU-Net successfully identified single tumor masses in LA phase, matching clinical corrections, PV phase segmentations showed disconnected regions in two subjects (9.5% of cases), presenting two and three separate masses respectively. These findings support the network's reduced accuracy in tumor delineation in phases other than LA, where contrast characteristics are less optimal for PDAC visualization. However, a good segmentation of the tumor also in other phases could be relevant to identify PDAC position more precisely. Indeed, positional information can be later used during Endoscopic Ultrasonography which is performed in combination with fine needle aspiration to obtain PDAC sample biopsy [14].
Our registration methodology was developed to address the

fundamental challenge of multi-phase PDAC imaging analysis: optimizing tumor segmentation across phases with varying contrast characteristics. A comprehensive statistical evaluation demonstrated the robustness of our registration approach. Although comparison of DSC between registered PV phase and clinically-corrected LA phase showed significant differences, the improvement in the median value relative to nnU-Net PV is evident in Figure 2 highlighting the positive impact of registration. In addition, in terms of centroid distances, the registered PV did not show significant differences compared to nnU-Net LA, whereas nnU-Net PV segmentation differed significantly from the LA phase. Therefore, the PDAC DSC values obtained through our co-registration approach can be considered valid and clinically meaningful. While our median DSC of 0.66 is lower than the 0.72 reported by Cao et al. [16], it should be noted that this comparison is not with our final intended outcome. Our registration method aims to obtain PDAC segmentation on the venous phase, but we currently lack the clinically correct venous phase mask for direct validation. Therefore, this comparison represents an intermediate step rather than our ultimate goal, and we expect higher DSC values when starting from clinically correct venous phase segmentations. Nevertheless, our approach offers significant advantages in terms of clinical deployability, operating without GPU requirements unlike GPU-dependent methods such as those employed by Cao et al. or VoxelMorph++ [13], thereby ensuring broader accessibility in routine clinical environments. The patient-specific method selection strategy proved crucial for optimizing registration performance, as evidenced by the distinct patterns observed across our cohort. The superior performance of Elastic Focused Pancreas registration with phase-aware enhancement (M6) in the largest subset of patients (33.3%) underscores the value of combining non-linear transformation capabilities with clinically-informed enhancement patterns. This finding aligns with the inherent complexity of pancreatic anatomy and the significant tissue deformation that can occur between contrast phases in PDAC patients. The complete absence of optimal performance for multi-step rigid-to-elastic approaches (M2 and M5) warrants careful consideration. This unexpected finding suggests that sequential registration strategies may introduce cumulative errors that outweigh their theoretical advantages. The consistent benefit of phase-aware enhancement across different transformation types (appearing in 12/21 optimal selections) validates our hypothesis that incorporating contrast enhancement patterns improves registration accuracy. By weighting the mutual information metric according to clinically established HU thresholds for PDAC, the registration process can prioritize anatomically stable landmarks while de-emphasizing regions prone to contrast-related variations. The variability in optimal methods across patients highlights the heterogeneous nature of PDAC presentation and the importance of personalized computational approaches. Unlike many registration applications where a single method may suffice, our results demonstrate that PDAC patients exhibit sufficient anatomical and pathological diversity to benefit from method selection strategies.

## V. Conclusions

### A. *Main Findings*

Our registration methodology successfully addresses the fundamental challenge of multi-phase PDAC imaging analysis by optimizing tumor segmentation across phases with varying contrast characteristics. The patient-specific method selection strategy proved crucial, with Elastic Focused Pancreas registration with phase-aware enhancement (M6) demonstrating superior performance in 33.3% of patients. The consistent benefit of phase-aware enhancement across different transformation types validates our hypothesis that incorporating contrast enhancement patterns improves registration accuracy. Our patient-specific method selection ensures that at least one registration approach achieves clinically acceptable performance (DSC > 0.70) for each patient in our cohort, providing robust coverage across diverse anatomical and pathological presentations encountered in PDAC patients. Our approach enables efficient multi-phase PDAC analysis by requiring manual correction only in the optimal LA phase, potentially improving diagnostic efficiency and treatment planning.

### B. *Limitations*

Several limitations should be acknowledged. First, the relatively small sample size of 21 patients represents a constraint that limits the generalizability of our findings. Second, our methodology was applied exclusively to the PV phase, which may limit its broader applicability. Additionally, the phase-aware enhancement thresholds were derived from literature and may not be optimal for all scanner types or contrast protocols. The computational complexity of evaluating six methods per patient (requiring 60+ minutes total) may also limit immediate clinical applicability.

### C. *Future Directions*

Future work should focus on developing predictive models for optimal method selection based on patient characteristics to reduce computational time from 60+ minutes to approximately 10 minutes for routine clinical use. Expanding the patient cohort would enable more robust statistical analyses and strengthen the reliability of the approach. Extending this methodology to additional CT phases, such as the delayed phase, could enhance tumor detection capabilities, particularly for small PDACs exhibiting isoattenuation in LA phase [15].

## Acknowledgment

This research is supported by the PNRR Project PNRR-MAD-2022-12376716.

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
