# OpenReview forum: "Patient specific Pancreatic Ductal Adenocarcinoma segmentation in multiphase CTs through a registration methodology"
_IEEE.org/EMBS/BHI/2025/Conference — BHI 2025_

### Official Review · Reviewer_Q5az · 2025-07-01
**Patient specific Pancreatic Ductal Adenocarcinoma segmentation in multiphase CTs through a registration methodology**

**Confidence:** 4
**Clarity Of Writing:** good
**Clinical Significance:** good
**Methodological Novelty:** good
**Overall Rating:** 7

**Experiments And Results:**

fair

**Questions For The Authors:**

Why focus exclusively on pancreas registration when PDAC segmentation is the clinical goal? What is the tumor registration accuracy specifically?
How long does the registration process take per patient? Is running six methods and selecting best one clinically feasible?
What happens when all six methods perform poorly? Do you have criteria for identifying registration failure?

**Strengths:**

Addresses real challenge in PDAC imaging where different contrast phases provide complementary information but with varying visualization quality.
 By requiring manual correction only in optimal LA phase and automatically transferring to PV phase, this could save significant clinical time.
Recognizing that no single registration method works best for all patients and implementing personalized method selection is sophisticated approach.
 Incorporating tissue enhancement patterns (hypovascular PDAC, arterial structures, normal parenchyma) into registration through weighted mutual information is clever technical contribution.
Testing six different registration strategies with proper parameter optimization shows thorough methodology.
Using radiologist-corrected segmentations as ground truth provides strong clinical relevance.

**Summary Of The Paper:**

This manuscript presents a registration-based methodology for improving pancreatic ductal adenocarcinoma (PDAC) segmentation across multiple CT phases. The authors utilize nnU-Net to generate automatic segmentations for both late arterial (LA) and portal venous (PV) phases in 21 PDAC patients. They observe that LA phase provides better visualization and segmentation accuracy for PDAC compared to PV phase. To address this, they propose six different registration strategies (three standard and three phase-aware) to align PV phase segmentations with clinically-corrected LA phase segmentations. The best-performing method for each patient is selected based on Dice similarity coefficient (DSC). Their approach achieves median DSC of 0.85 for pancreas segmentation in registered PV phase, compared to 0.79 for unregistered PV phase and 0.90 for LA phase.

**Weaknesses:**

Authors don't compare their approach against other multi-phase registration techniques or commercial software.
While they mention tumor segmentation performance, the focus shifts entirely to pancreas registration without fully addressing tumor registration accuracy.
No information about computation time, hardware requirements, or feasibility for clinical deployment.

---

### Official Review · Reviewer_jzkY · 2025-07-17
**A good conference paper**

**Confidence:** 3
**Clarity Of Writing:** great
**Clinical Significance:** great
**Methodological Novelty:** great
**Overall Rating:** 7
**Final Rating:** 8

**Experiments And Results:**

great

**Questions For The Authors:**

I have no questions for the authors that I would like to ask.

**Strengths:**

1. The language used is clear and concise.
2. The paper is supported with adequate tables and figures.
3. Methodology followed, results acquired and conclusions drawn are sufficient.

**Summary Of The Paper:**

In this study, authors present a patient-specific paradigm used clinically validated late arterial (LA) phase
segmentation and interphase registration to align portal venous (PV) phase segmentations with the corrected LA phase which enables more accurate PV phase annotation. Authors claim that their methodology addresses phase-dependent segmentation challenges while optimizing clinical workflow which potentially improves diagnostic efficiency and treatment planning.

**Weaknesses:**

1. Long paragraphs can be broken down into individual ones without breaking the flow of paper to increase its readability
For example, in the conclusion section, the future directions can be its own paragraph.
2. A comparison with the literature would be nice preferably in a tabulated form.

---

### Official Review · Reviewer_DsKM · 2025-07-17
**Patient specific Pancreatic Ductal Adenocarcinoma segmentation in multiphase CTs through a registration methodology**

**Confidence:** 3
**Clarity Of Writing:** good
**Clinical Significance:** good
**Methodological Novelty:** good
**Overall Rating:** 7

**Experiments And Results:**

good

**Questions For The Authors:**

No questions for the authors

**Strengths:**

-

**Summary Of The Paper:**

This paper presents a patient-specific registration methodology aimed at improving automated segmentation of Pancreatic Ductal Adenocarcinoma (PDAC) in multiphase CT imaging. The authors propose a nnU-Net to generate baseline segmentations, followed by deformable registration of portal venous (PV) phase data to late arterial (LA) segmentations. The study seems very interesting, the methodologies and results are adequately reported and explained.

**Weaknesses:**

-